# Percutaneous Endoscopic Transforaminal Lumbar Interbody Fusion (PETLIF): Current Techniques, Clinical Outcomes, and Narrative Review

**DOI:** 10.3390/jcm12165391

**Published:** 2023-08-19

**Authors:** Koichiro Ono, Daisuke Fukuhara, Ken Nagahama, Yuichiro Abe, Kenji Takahashi, Tokifumi Majima

**Affiliations:** 1Department of Orthopedic Surgery, Nippon Medical School, 1-1-5 Sendagi, Bunkyo-ku, Tokyo 113-8603, Japan; d-fukuhara@nms.ac.jp (D.F.); t-majima@nms.ac.jp (T.M.); 2Department of Orthopedic Surgery, Kyoto Prefectural University of Medicine, 465 Kajii-cho, Kamigyo-ku, Kyoto 602-8566, Japan; t-keji@mbox.kyoto-inet.or.jp; 3Sapporo Endoscopic Spine Surgery Clinic, 1-13, North-16, East-16, Higashi-ku, Sapporo 065-0016, Japan; k.nagahama.8@gmail.com; 4Sapporo Medical Research, 1-20-1501, Kita13 Higashi2, Hidashi-ku, Sapporo 065-0013, Japan; menchixp@gmail.com

**Keywords:** percutaneous endoscopic transforaminal lumbar interbody fusion, PETLIF, percutaneous endoscopy, full-endoscopy, lumbar interbody fusion, minimally invasive

## Abstract

Full endoscopic techniques are becoming more popular for degenerative lumbar pathologies. Percutaneous endoscopic lumbar interbody fusion (PETLIF) is a minimally invasive surgical technique for spondylolisthesis and lumbar spinal canal stenosis with instability. Nagahama first introduced PETLIF in 2019. This study investigated the clinical outcomes and complications of 24 patients who underwent PETLIF in our facility and compared them with previous studies. Literature searches were conducted on PubMed and Web of Science. The PETLIF surgical technique involves three steps to acquire disc height under general anesthesia. The procedure includes bone harvesting, spondylolisthesis reduction, endoscopic foraminoplasty, disc height expansion using an oval dilator, and intervertebral disc curettage. A cage filled with autologous bone is inserted into the disc space and secured with posterior fixation. Patients underwent PETLIF with an average operation time of 130.8 min and a blood loss of 24.0 mL. Postoperative hospital stays were 9.5 days. Improvement in VAS, disc height, spinal canal area, and % slip was observed, while lumbar lordosis remained unchanged. Complications included end plate injury, subsidence, and exiting nerve root injury. The differences between PETLIF and the extracted literature were found in patients’ age, direct decompression, epidural or local anesthesia, approach, order of PPS, and cage insertion. In conclusion, PETLIF surgery is a practical, minimally invasive surgical technique for patients with lumbar degenerative diseases suffering from back and leg pain, demonstrating significant improvements in pain scores. However, it is essential to carefully consider the potential complications and continue to refine the surgical technique further to enhance the safety and efficacy of this procedure.

## 1. Introduction

Patients with lumbar spinal canal stenosis (LSCS) suffer from low back pain and leg pain caused by diminished space available for the neural elements [1]. LSCS results in disability and reduces the quality of life, morbidity, and function. Conservative treatment, in the form of medication and rest, is generally the first choice for this pathology. However, patients with intolerable pain for whom conventional treatment fails or who develop paralysis of limb muscles are considered for surgical intervention. LSCS has become the most frequently encountered pathological condition in orthopedic practice and the most common reason for spine surgery over 65 years old [2].

Lumbar interbody fusions are one of the surgical options for LSCS patients who require spinal stabilization and correct deformity. Conventional posterior lumbar interbody fusion (PLIF) or transforaminal lumbar interbody fusion (TLIF) techniques have been used as effective surgical methods for degenerative lumbar spine diseases [3]. However, these posterior approach techniques are associated with a certain amount of paraspinal muscle damage, which may cause failed back syndrome [4]. Therefore, minimally invasive lumbar interbody fusion (MIS-TLIF) using percutaneous pedicle screwing (PPS) is widely used to reduce the damage to paraspinal muscles [5]. The other concerns for the posterior approaches which need to be considered during surgery are direct decompression of neural elements, incidental durotomy, blood loss, and epidural hematoma. Recently, lateral lumbar interbody fusion (LLIF) has gained popularity, providing indirect decompression for LSCS without damage to posterior elements [6].

Moreover, indirect decompression can avoid those concerns of direct decompression. Compared to MIS-TLIF, LLIF has reduced blood loss, shorter operation time, and improved low back pain [7]. However, multiple neurological and other injuries have been observed after LLIF [8]. Some are life-threatening, including major vascular injuries and bowel perforations [8].

Endoscopic lumbar interbody fusion (ELIF) is an increasingly popular technique for treating lumbar degenerative disease with minimal invasion [9,10,11,12,13]. Percutaneous endoscopic transforaminal lumbar interbody fusion (PETLIF), one of the ELIFs recently introduced by Nagahama [14], is a minimally invasive interbody fusion technique that provides indirect decompression. One of the novelties of the PETLIF is the application of an entire endoscope to the lumbar interbody fusion using a posterolateral approach to the intervertebral disc via Kambin’s triangle for the passage of an interbody cage. The other important aspect of PETLIF is the indirect decompression technique using an oval dilator and sleeve [14]. Disc height is expanded by rotating the oval dilator and sleeve, and this maneuver is critical for acquiring indirect decompression [14]. The advantage of PETLIF over other minimally invasive techniques is that indirect decompression can be achieved without concern about life-threatening complications. Here, the updated techniques and clinical outcomes of PETLIF are introduced and compared to previous reports.

## 2. Materials and Methods

### 2.1. Literature Search

To compare our techniques and clinical outcomes with other lumbar interbody fusion using endoscopy, a PubMed and Web of Science literature search was performed using the following search terms: (full-endoscopic lumbar interbody fusion) OR (percutaneous endoscopic lumbar interbody fusion). The database search was performed on 24 January 2023. All titles, abstracts, and the full text of relevant articles were reviewed. Studies regarding non-English-language articles, animal or cell experiments, duplicated studies or patients, case reports, reviews, letters, comments, technical notes, and different surgery lacking the data for reviewing and a follow-up period of less than one year were excluded. The number of articles included and excluded is shown in a flow chart. As shown in Figure 1, the initial literature search resulted in 160 articles in PubMed and 73 articles in Web of Science. Fifty-eight duplicates were identified, and based on the and exclusion criteria, 16 studies were extracted for review (Figure 1).

### 2.2. Data Extraction from the Manuscripts

The following data were extracted from the manuscripts: (1) author, journal, and year, (2) sample size, (3) patients’ age, (4) follow-up periods, (5) anesthesia: local or general, (6) approach: transforaminal or interlaminar, (7) direct or indirect decompression, (8) order of PPS and cage insertion, (9) postoperative hospital days, (10) clinical outcomes and complications (Table 1).

### 2.3. Patient Population

From February 2021 to December 2021, 24 patients who had undergone PETLIF for degenerative lumbar pathologies were included in this study (Table 2). The inclusion criteria were as follows: (1) single-segment LSCS or lumbar disc herniation with segmental instability; (2) Grade 1 or 2 degenerative spondylolisthesis; (3) failure of conservative treatment for more than three months; (4) follow-up periods more than six months after the surgery. The exclusion criteria were as follows: (1) Grade 3 or above degenerative spondylolisthesis; (2) history of lumbar surgery; (3) multi-level lesion; (4) direct decompression case; (5) follow-up periods less than 12 months. Lumbar instability was defined as progression of slippage greater than 3 mm, posterior opening over 5°, or segmental angle changes more than 10° on flexion and extension observed in lateral X-rays [15,16].

Appropriate preoperative assessments were conducted for all patients before the operation. Twenty-four patients were operated on at one intervertebral level using the PETLIF technique by a single surgeon (KO) (Table 3). There were 13 males and 11 females, and the average age was 70.5 ± 2.2 years. Fourteen patients were operated on in the L4/5 level, nine in L3/4, and one in L2/3 (Table 2). Operation time and intraoperative blood loss were recorded. The visual analog scale (VAS) for back and leg pain and the Japanese Orthopedic Association (JOA) score were observed at one month, three months, six months, and final follow-up (Table 4). Intervertebral disc height, an area in the spinal canal, and % slip were measured (Table 5). Complications were also explored in an electronic medical chart (Table 3).

### 2.4. Quality Assessment

Two authors completed the data quality assessment independently (KO, DF). The following variables were extracted for the included studies: first author name, published year, study design, disease type, surgical technique, approach, sample size, and age. The clinical variables were operative time, anesthesia, estimated blood loss (EBL), post-op hospital stay, follow-up periods, direct or indirect decompression, and order of the PPS and cage insertion. Discussion between the two authors resolved any disagreement during data extraction and quality assessment.

### 2.5. Surgical Technique of PETLIF

#### 2.5.1. Preparation and Planning

The PETLIF procedure was updated from the original technique [14]. There are three steps to acquiring disc height during the process. All PETLIFs were performed under general anesthesia. The somatosensory-evoked potentials, electromyography, and free-run electromyography were monitored throughout the procedure using a nerve monitoring system (NVM5^®^, NuVasive, San Diego, CA, USA). The patients were in a prone position on the operating table (Figure 2). The primary surgeon usually stands on the patient’s left side. In scoliosis cases, the approach is from the concave side to dissociate rigid scoliosis. On the surgeon’s side, from the cranial to the caudal, the order is fluoroscopy (F), the primary surgeon (S), the table (T), and the nurse (Ns), whereas on the opposite side of the surgeon there is fluoroscopy (F), endoscopic monitor (EM), and fluoroscopic monitor (FM). Behind the EM and FM, a platform for hanging irrigation water (water), a high-speed drill, and a bipolar console (console) is placed (Figure 2). According to the C-arm fluoroscope, percutaneous pedicle screw (PPS) entry points, approximately 1cm from the lateral edge of the pedicle, posterior-lateral entry for the cage, and incision for bone harvesting were marked on the skin (Figure 3a). Posterior-lateral entry was about 8–9 cm from the midline of the disc level to acquire 45°, measured by a CT scan before the surgery. After marking, the operation site was sterilized, and surgery was started.

#### 2.5.2. Bone Harvesting

Cancellus bone from the ilium was harvested for grafting. A 3 cm skin incision was made along the iliac crest from 2 fingers posterior to the superior anterior iliac spine, and the iliac crest was developed under the periosteum. The medial aspect of the iliac crest was opened with a chisel, and cancellous bone was harvested with a bone curette and preserved for bone grafting.

#### 2.5.3. Reduction of Spondylolisthesis Using PPS

Pedicle screws were inserted percutaneously under fluoroscopic guidance using the Pisces Spinal System^®^ (Japan Medical Dynamic Marketing, Tokyo, Japan) (Figure 3b). The screws’ diameter and length were determined by preoperative CT scan. Rod length was selected, and two rods were inserted bilaterally. The caudal set screws were fixed with the cranial side of the rods kept a certain distance from the screw head (Figure 3c). The spondylolisthesis of the cranial vertebra was then corrected by tightening the set screws (Figure 3d).

#### 2.5.4. Expansion of Intervertebral Foramen

The VERTEBRIS lumbar instrument set (RIWOspine GmbH, Knittlingen, Germany) was used throughout the endoscopic procedure. After the spondylolisthesis reduction, a pencil dilator was inserted from the posterior-lateral entry to the lateral facet joint. The pencil dilator tip was then slid into the foramen from the ventral aspect of the facet joint. When the tip of the dilator could pass through the foramen, expansion of the foramen was unnecessary. Otherwise, the expansion of the intervertebral foramen was performed. An 8 mm sleeve and rigid endoscope were inserted through the dilator. Soft tissue was removed using forceps and a bipolar probe (Trigger-Flex Bipolar System; Elliquence, New York, NY, USA) under endoscopic control (Figure 4a), and the lateral aspect of the facet joint was exposed. Subsequently, the ventral part of the foramen was drilled using a high-speed burr (Nakanishi Primado2^®^ total surgical system, Tochigi, Japan) (Figure 4b,c) until the tip of the dilator was able to pass through the foramen (Figure 5a,b).

#### 2.5.5. The First Step for Acquiring the Disc Height

At this point, the tip of the dilator is supposed to locate at the posterior intervertebral disc and medial line of the pedicle. The cranial set screws were freed, and the dilator was hammered into the disc space at an angle of 45° from a horizontal line. (Figure 5c) Expansion of the disc height was observed during the dilator insertion into the disc space, and the set screws were tightened to maintain the acquired disc height.

#### 2.5.6. The Second Step for Acquiring the Disc Height

The oval dilator (9 × 7 mm), oval sleeve (10.6 × 9 mm), U-shaped oval sleeve, dedicated cobb elevator, ring curette, and J-shaped nerve retractors (Japan Medical Dynamic Marketing, Tokyo, Japan) were newly developed or updated and used from this step. The pencil dilator was removed, and the oval dilator was inserted through the same passage with a short axis craniocaudally. (Figure 5d) The cranial set screws were freed, and the oval dilator was rotated 90 degrees, so that further expansion of the disc height was acquired with the long axis craniocaudally (Figure 5e). Set screws were tightened to maintain the disc height.

#### 2.5.7. The Third Step for Acquiring the Disc Height

The oval dilator was rotated and the short axis was set craniocaudally. The oval sleeve was then inserted over the oval dilator. The cranial or caudal set screws were freed, and the oval dilator was rotated 90 degrees; that final expansion of the disc height was acquired with long axis craniocaudally (Figure 5f). Set screws were tightened to maintain the disc height. Theoretically, after the third step the disc height increased by at least 10.6 mm.

#### 2.5.8. Intervertebral Disk Curettage

The oval sleeve was set backward, and the disc curettage was performed through the sleeve fluoroscopically (Figure 4d) or endoscopically (Figure 5g,h). In the endoscopic view, the disc was curetted to the extent that the bony endplate was exposed (Figure 4e).

#### 2.5.9. Cage Insertion

The oval sleeve was changed to the U-shaped sleeve and the cage trial was inserted, starting at 8 mm and expanding to an appropriate size, usually 9–11 mm. Once the size was determined, pairs of J-shaped nerve retractors were set through the U-shaped sleeve (Figure 6a). Finally, the cage (Vusion3DTi, Japan Medical Dynamic Marketing), filled with autologous grafted bone from the ilium or spinal process, was driven toward the disc space while placed between the retractors (Figure 6b). The sleeve was removed when the cage passed the exiting nerve root, and the cage was driven into the disc space (Figure 6c). The cage was placed on a mechanically robust rim to avoid subsidence (Figure 6d).

#### 2.5.10. End of the Surgery

The surgery was terminated with the final fastening of the set screw for posterior fixation.

### 2.6. Statistical Analysis

The data were analyzed using chi-square and Fisher’s exact test for nominal data and an independent *t*-test in continuous data. A statistically significant difference was determined when *p* < 0.05.

## 3. Clinical Data of the Previous Literature (Table 1)

(1) ELIF is particularly popular in countries with advanced healthcare systems and a strong focus on minimally invasive surgical techniques, including the United States [17], Germany, South Korea [18], Japan [14,19], and China [12]. (2) Sample sizes ranged from 7 [20] to 114 [21] cases, an average of 54.7 cases. (3) Mean age of included cases in the literature ranged from 36.1 [22] to 70.7 [23] years old. (4) Follow-up periods were 12 [24] to 24.7 [25] months. (5) General anesthesia was applied in most of the manuscripts, identical to our cases. In contrast, epidural [20,21,26] or local [20,21,23,27] anesthesia was performed in some reports. (6) Two significant approaches, transforaminal or interlaminar, were utilized to install the intervertebral cage. Among sixteen reports, the interlaminar approach was used in six papers, transforaminal in eleven, and one report used two approaches. (7) In most cases, direct decompression was performed. In contrast, indirect decompression has been applied some cases in the transforaminal approach [14,22,27,28]. (8) PPS is installed after cage insertion in most of the manuscripts other than PETLIF [14] and PE-TLIF recently published [23]. (9) Postoperative hospital days ranged from 3.11 [29] to 8.87 [30] days. (10) Operation time differed among papers from 87.5 [31] to 355 [22] min. A shorter operation time in ELIF compared to other procedures was reported in one report [30] and longer [25] in two papers. Radiation time also differed, in that a shorter time of ELIF was reported in one paper [28] and longer in the other [24]. ELIF had less surgical trauma [24], less postoperative low back pain [23], less estimated blood loss [23] and shorter hospital stay [23] compared to the other surgical techniques.

## 4. Results

### 4.1. Patients’ Characteristics and Clinical Outcomes

All patients underwent PETLIF surgery for their pathology (Table 3). The mean operation time was 130.8 ± 9.22 min (82–260 min). The average estimated intraoperative blood loss was 24.0 ± 4.9 mL (7–97 mL) (Table 2). On average, patients stayed in the hospital for 9.4 ± 0.6 days (5–21 days) after the surgery (Table 2). The average follow-up period was 20.7 ± 0.5 months (Table 2). The JOA score, VAS for back and leg pain at one month, three months, and six months after the operation and the final follow-up improved significantly compared to those before the procedure (*p* < 0.05) (Table 4).

The mean preoperative intervertebral disc height of 5.1 mm improved to 8.0 mm postoperatively and was maintained at 7.3 cm at the final follow-up, the spinal canal area from 41.8 mm^2^ to 108.6 mm^2^, and % slip improved from 17.3% to 4.5%. On the other hand, lumbar lordosis (LL) averaged 33.3° preoperatively, 32.6° postoperatively, and 33.3° at the final follow-up. As a result, the mean pelvic incidence (PI) minus LL (PI-LL) was 15.6° preoperatively, 16.3° postoperatively, and 15.6° at the final follow-up, with no improvement as well (Table 4). Bone fusion was observed in 21 cases out of 24 patients, and the rate was 87.5% (Table 3). The three cases in which bone fusion has not been achieved are still under outpatient observation while continuing to receive teriparatide because their pain is under control.

### 4.2. Complications

End plate injury was observed in one case, subsidence in seven patients (29.2%), and exiting nerve root injury in cases 10 and 20 (8.3%). (Table 3). The recovery time was 4 months in case 10 and 5 months in case 20. In case 10, the patient experienced lower limb fatigue on the cage entry side after walking long distances, and the hospital stay was extended to 21 days after the surgery. In case 20, muscle strength was affected, such as the disability to ascend and descend stairs, but the patient was discharged from the hospital on post-operation day 8. These symptoms improved along with the pain.

### 4.3. Representative Cases

Case 1: A 70-year-old woman was diagnosed with degenerative spondylolisthesis of the L4 vertebra with instability by X-ray, CT, and MRI. VAS for back and leg pain were 100 and 60 mm, and the JOA score was 18/29. The spinal canal area was 95 mm^2^ at L4/5 disc level, and the % slip was 21% (Figure 7a–d). After the failure of conservative treatment, PETLIF was performed. One month after surgery, VAS back and leg pain improved significantly to 0 and 10 mm. Postoperative X-ray and MRI showed that the % slip was 0%, and the spinal canal area was expanded to 153 mm^2^ (Figure 7e–h). Bone fusion of L4/5 was confirmed by X-ray and CT scan one year after the surgery (Figure 8a–d).

Case 2: A 63-year-old woman was diagnosed with degenerative spondylolisthesis of the L4 vertebra with instability by X-ray, CT, and MRI. VAS for back pain was 10 mm and leg pain 40 mm, and the JOA score was 21/29. After the failure of conservative treatment, PETLIF was performed. One month after surgery, the VAS of leg pain improved significantly to 0 mm, whereas back pain was 10 mm. Postoperative X-ray and MRI showed spondylolisthesis was well reduced comparable to case 1 (Figure 9e–h). 

## 5. Discussion

PETLIF is a minimally invasive surgical procedure for degenerative lumbar pathologies, such as degenerative disc disease, spondylolisthesis, and spinal canal stenosis. The critical concepts of the PETLIF procedure are described below.


Obtaining indirect decompression with reduced lumbar spondylolisthesis and disc height using PPS first and then an oval-shaped sleeve and dilator placed by an endoscopic transforaminal approach [14].Curettage of the affected intervertebral disc is performed under fluoroscopic and endoscopic guidance using special instruments dedicated to PETLIF [14].


Lateral lumbar interbody fusion (LLIF), such as extreme lateral lumbar interbody fusion (XLIF) and oblique lateral lumbar interbody fusion (OLIF), is the technique for obtaining indirect decompression techniques that have several advantages over direct decompression and have become more often used for lumbar degenerative pathologies [32]. These include faster recovery times, reduced blood loss, less infection, and less scarring due to the procedure’s minimally invasive nature [33]. While the techniques offer these benefits, there are potential risks of life-threatening complications such as great vessel injury and intestinal injury [32]. PETLIF provides this indirect decompression with the least risk of these life-threatening complications [14]. Most of the selected articles on endoscopic lumbar interbody fusion techniques used direct decompression, while only two papers, including PETLIF, performed indirect decompression [34]. Indirect decompression was achieved by an expandable cage in the other non-PETLIF paper [28]. Another article did not perform direct decompression, but it was not a technique for obtaining indirect decompression such as expanding disc height [22] (Table 1). In this regard, PETLIF is the only procedure that aims to bring indirect decompression using an oval dilator, which is very different from other endoscopic minimally invasive lumbar interbody fusion procedures [14]. PETLIF can be particularly powerful at the L5/S1 level. The risk of vascular injury is very high at the L5/S1 level because the descending aorta and inferior vena cava are bifurcated and located on the trajectory of the lateral and anterior approaches [35], so PETLIF, which can obtain indirect decompression via this intervertebral foramen, has the potential to be a safe and effective alternative technique for the L5/S1 level.

Another critical aspect of the PETLIF obtaining indirect decompression is that PPS is placed before the cage to avoid endplate injury during cage insertion [14]. In the three steps of acquisition of disc height, the PPS is fastened each time it is lifted to maintain disc height and avoid endplate injury. PETLIF was the only technique among the 16 extracted articles that inserted the PPS before cage installation (Table 1). Although cage subsidence was observed in four of the twenty-four cases (Table 4), all were early cases, and cage subsidence was not observed in the later patients in which the surgical steps were performed correctly, suggesting that endplate damage could have been avoided by inserting the cage while maintaining disc height using PPS.

Endplate preparation is performed under endoscopic assistance during PETLIF, allowing visualization of the disc space and safe curettage of the disc to the very edge of the subchondral bone. This ideal disc space preparation allows for a high probability of bone fusion [36]. Our series showed a 95% fusion rate at one year postoperatively, indicating that endoscopic curettage of the disc may have resulted in a more secure bony fusion (Table 2).

Since PETLIF is a technique that expects indirect decompression by expanding the disc height, PETLIF has the following contraindications. (1) Stiff segments where an expansion of disc height is not expected. (2) Calcification of the ligamentum flavum where indirect decompression is not expected. (3) Severe disc space narrowing [11], which may cause an endplate injury while inserting a dilator. (4) High-grade spondylolisthesis (grade > 2) [18] that reduction by PPS cannot be expected. (5) Poor bone quality that reduction and acquisition of disc height are also not expected [11,37]. For these cases, MIS-TLIF [38], XLIF [6], or conventional open surgery can be the alternative techniques to expect favorable clinical outcomes.

Significant improvements in VAS scores for back and leg pain were observed at one month, three months, and six months after the operation and the final follow-up in 24 cases (Table 3). In addition, less postoperative low back pain compared to other techniques was observed [21]. The reason for this reduced pain may be due to less surgical trauma [24], less estimated blood loss [29], and shorter hospital stay [39] with ELIF compared to the other surgical techniques.

Most papers reported favorable clinical outcomes with ELIF identical to other techniques, whether conventional or MIS-TLIF. (Figure 2). The included studies had an average follow-up period of 20.7 months. As ELIF is a relatively new technique, long-term results will be investigated in the future. Identical good clinical outcomes have been observed over long periods, and ELIF can be a valid alternative to the other techniques.

The mean age of our patients was higher than that of the cases included in the extracted papers (70.5 vs. 36.1 [22] to 65.02 [31]). The indication of lumbar interbody fusion in young patients is avoided in our facility due to the potential for permanent restrictions and loss of flexibility in the spine [40].

In our cases, all PETLIFs were performed under general anesthesia, as was the case in most of the manuscripts. In contrast, epidural anesthesia [17] was performed in some reports. If PETLIF can also be performed under local anesthesia, lumbar interbody fusion can be an option for patients with poor general conditions who have difficulty undergoing general anesthesia [20].

Postoperative hospital days were 9.46 days in our cases, which was longer than the extracted papers (3.11 [29] to 8.87 [30] days). Since our patients were older than those in the documents as described above, patients may have needed more time for rehabilitation. Additionally, differences in social backgrounds and the healthcare system in Japan may have influenced the length of hospitalization.

The procedure in our series is relatively quick and minimally invasive compared to the included studies, with a mean operation time of 130.8 min and an average intraoperative blood loss of 24.0 mL (Table 1). PETLIF’s nature is indirect decompression, which might shorten the operation time. Moreover, less blood loss may represent the minimally invasive nature of PETLIF.

Despite the overall success of the procedure, some complications were observed in a subset of patients (Table 4). End plate injury occurred in one case, subsidence in four cases, and exiting nerve root symptoms in five cases. Most of these complications, except one exiting nerve root injury, occurred in the early cases after introduction. This may be due to the inaccurate performance of the surgical steps during the procedure or inadequate foraminoplasty. These complications highlight the need for careful patient selection and proper surgical steps to minimize potential risks.

The increase in intervertebral disc height and improvement in % slip resulted in the enlargement of the spinal canal area, further supporting the efficacy of PETLIF surgery in addressing the underlying structural issues. On the other hand, LL did not improve in our series, and a good clinical outcome requires the achievement of proper spinopelvic alignment [41], and PETLIF has room for improvement in this regard in terms of proper sagittal alignment.

## 6. Conclusions

In conclusion, intervertebral PETLIF surgery is a practical, minimally invasive surgical technique for patients with lumbar degenerative diseases suffering from back and leg pain, demonstrating significant improvements in pain scores. However, it is essential to carefully consider the potential complications and continue to refine the surgical technique further to enhance the safety and efficacy of this procedure.

## Figures and Tables

**Figure 1 jcm-12-05391-f001:**
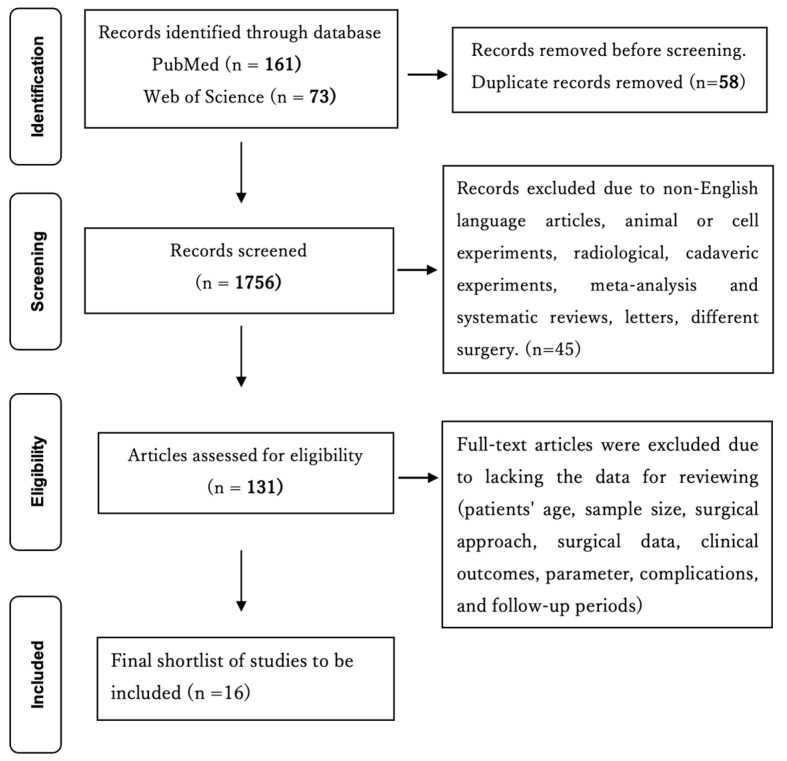
The flowchart describes the stepwise identification of studies to meet the search criteria (broad criteria, then narrowed down).

**Figure 2 jcm-12-05391-f002:**
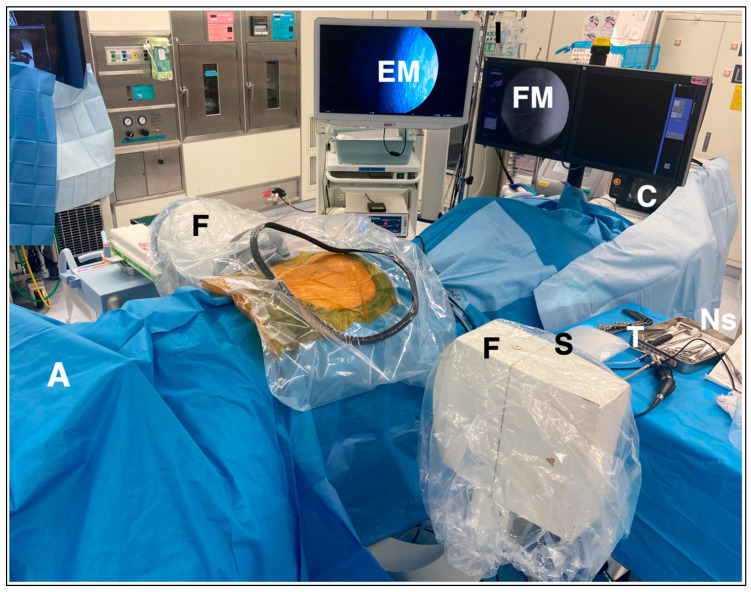
The primary surgeon (S) stands on the patient’s left side (standing on the right side, depending on the case). Anesthesiologist (A) stays at the cranial side of the patient, and the console (C) for electrocautery, bipolar and drill is located caudally. On the surgeon’s side, from the cranial to the caudal, the order is fluoroscopy (F), the primary surgeon (S), the table (T), and the nurse (Ns), whereas on the opposite side of the surgeon there is fluoroscopy (F), endoscopic monitor (EM), and fluoroscopic monitor (FM). Behind the EM and FM, a platform for hanging irrigation water (water), a high-speed drill, and a bipolar console (console) is placed.

**Figure 3 jcm-12-05391-f003:**
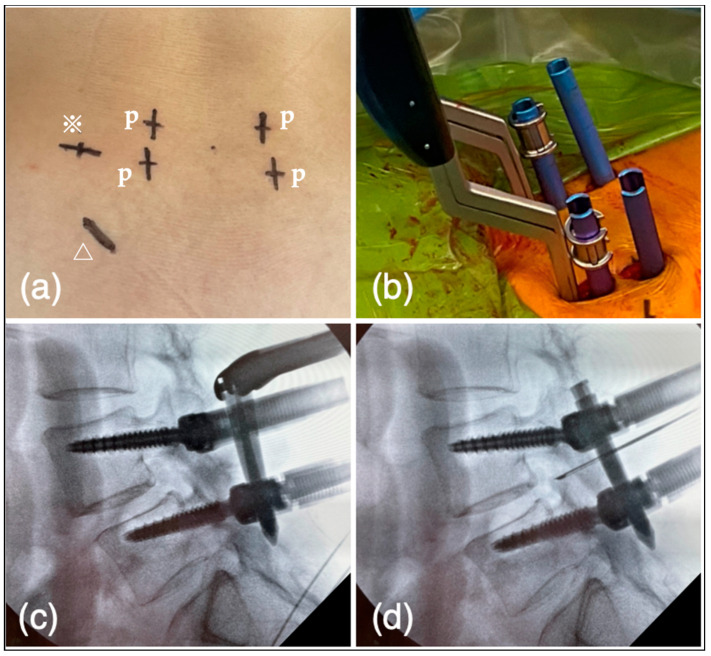
(**a**) According to the C-arm fluoroscope, PPS entry points (p), posterior-lateral entry for the cage (※), and incision for bone harvesting (△) were marked on the skin. (**b**) PPS was inserted under fluoroscopic guidance. (**c**) The caudal set screws were fixed with the cranial side of the rods kept a certain distance from the screw head. (**d**) The spondylolisthesis of the cranial vertebra was corrected by tightening the set screws.

**Figure 4 jcm-12-05391-f004:**
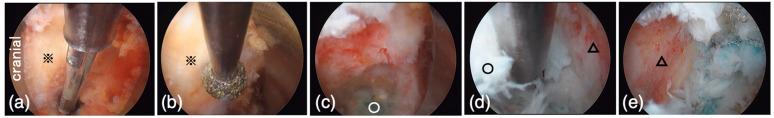
In endoscopic view during the expansion of intervertebral foramen. The left side is cranial in all figures. (**a**) The lateral aspect of the facet joint (※) was exposed after the removal of soft tissue using forceps and a bipolar probe. (**b**) The ventral part of the foramen (※) was drilled using a high-speed burr. (**c**) The blue-stained intervertebral disc (**○**) was observed after the expansion of the foramen. (**d**) The disc (**○**) curettage was performed. The bony endplate (**△**) at the caudal vertebrate was exposed. (**e**) Also, the bony endplate (**△**) at the cranial vertebrate was exposed.

**Figure 5 jcm-12-05391-f005:**
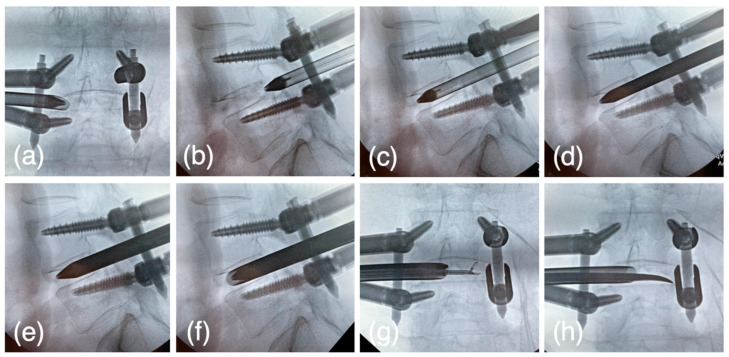
(**a**,**b**) The pencil dilator is inserted after the tip of the cannula reaches the medial line of the pedicle w/wo enlargement of the foramen. (**c**) 1st step: Loosen the set screw and drive the pencil dilator into the disc space. (**d**) Tighten the set screw and change to the oval dilator. (**e**) 2nd step: Loosen the set screw, 90° rotate the oval dilator to acquire the disc height. (**f**) 3rd step: Tighten the set screw and reverse the dilator, then insert the oval cannula over the dilator. Finally, loosen the set screw and the dilator is rotated with the cannula, and the disc height is expanded (**g**,**h**). The disc curettage was performed through the sleeve fluoroscopically or endoscopically using dedicated forceps and a Cobb elevator.

**Figure 6 jcm-12-05391-f006:**
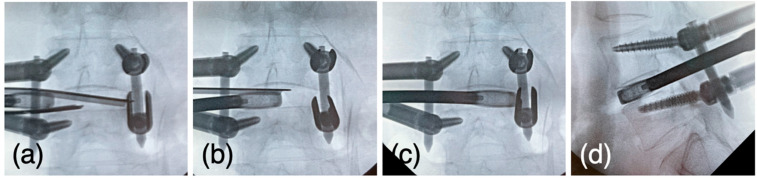
(**a**) The pairs of J-shaped nerve retractors were set through the U-shaped sleeve (**b**) The cage is driven toward the disc space while placed between the retractors. (**c**) The sleeve is removed when the cage passes the exiting nerve root, and the cage is driven to the contralateral border of the disc space. (**d**) The lateral view confirms the cage set on the rim.

**Figure 7 jcm-12-05391-f007:**
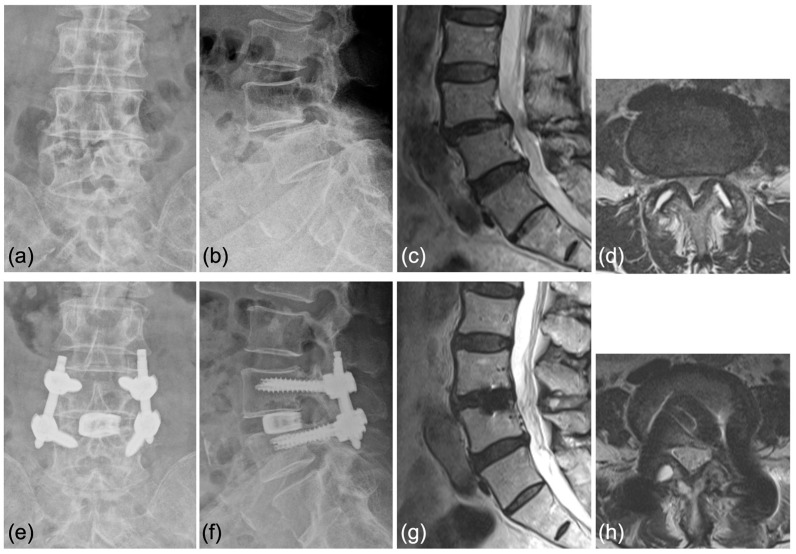
(**a**,**b**) Preoperative X-ray showing 2° L4 spondylolisthesis. (**c**,**d**) Preoperative MRI showed lumbar spinal stenosis at the L4/5 level. (**e**,**f**) Postoperative X-ray showed L4 spondylolisthesis was reduced to 0%. (**g**,**h**) Postoperative MRI showed the spinal canal area was expanded.

**Figure 8 jcm-12-05391-f008:**
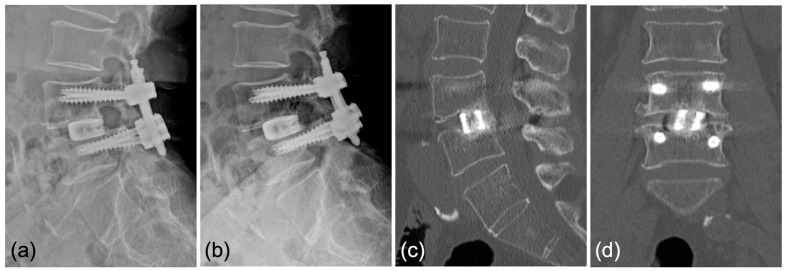
Bony fusion through the titanium cage was confirmed by (**a**,**b**) anterior and posterior flexed lateral X-rays and (**c**,**d**) sagittal and coronal CT scans.

**Figure 9 jcm-12-05391-f009:**
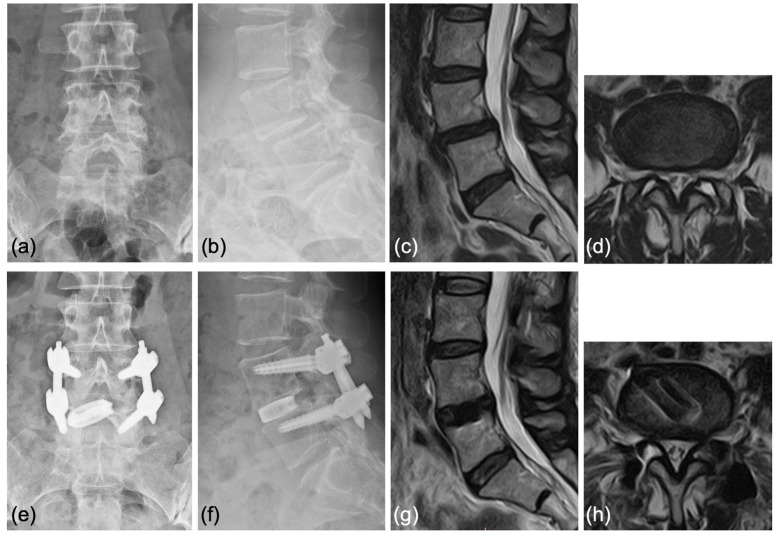
(**a**,**b**) Preoperative X-ray showing 2° L4 spondylolisthesis. (**c**,**d**) Preoperative MRI showed lumbar spinal stenosis at the L4/5 level. (**e**,**f**) Postoperative X-ray showed L4 spondylolisthesis was well reduced. (**g**,**h**) Postoperative MRI showed that the spinal canal area was expanded.

**Table 1 jcm-12-05391-t001:** Baseline Characteristics of Included Studies.

First Author	Year	Study Design	Disease Type	Surgical Technique	Approach	Sample Size	Age	Sex (M/F)	Op Time	Anesthesia	EBL	Post-Op Hospital Stay	Follow–Up Periods	Decompression (Direct/Indirect)	Order of PPS and Cage
Nagahama, K	2019 [14]	RS	DLS	PETLIF	TF	25	68.4	5/20	125.4	general	64.8	-	22.7	indirect	PPS → cage
Lin, L	2022 [15]	RS	LSCS	PE-LIF	IL	41	61.9	15/26	193.4	general	122.2	8.9	14.1	direct	cage → PPS
				MIS-TLIF	-	48	63.0	18/30	167.3	general	157.9	9.4	13.7	direct	
He, LM	2022 [16]	RS	DLS	PE-PLIF	IL	28	59.8	14/14	221.2	general	169.2	3.5	18.4	direct	cage → PPS
				Open PLIF	-	28	54.2	13/15	138.4	general	649.6	7.3	18.9	direct	
He, LM	2022 [17]	PS	LDH, DLS, LDD	PE-PLIF	IL	30	51.6	18/2	179.8	general	63.3	3.3	24.7	direct	cage → PPS
				Open PLIF	-	30	55.9	18/12	125.8	general	313.3	7.0	25.3	direct	
Wang, JC	2022 [18]	RS		TF	TF	14	35.0	3/11	121.8	local	-	-	>12	indirect	cage → PPS
				IL	IL	18	43.1	8/10	129.7	general	-	-	>12	direct	cage → PPS
Silva, AC	2022 [19]	RS		PELIF	TF	19	36.1	17/2	355	general	215.8	3.0	47	indirect	cage → PPS
Xue, Y	2021 [20]	RS		PETLIF	TF	41	46.3	11/9	140.3	general	65.6	2.4	16.1	direct	cage → PPS
				MISTLIF	-	48	47.1	12/8	170.6	general	140.5	4.5	15.8	direct	
Kim, HS	2021 [21]	RS		ETLIF(I)	TF	48	65.0	-	102.6	general	-	-	14.7	direct	cage → PPS
				ETLIF(O)	TF	38	68.4	-	87.5	general	-	-	11.6	direct	
He, L	2021 [22]	RS		PE-PLIF	IL	35	52.3	21/14	179.6	general	68.6	3.1	>12	direct	cage → PPS
Jiang, C	2021 [23]	RS	LSCS, DLS	Endo-PLIF	IL	24	59.5	10/14	209.2	epi	43.3	8.7	15.2	direct	cage → PPS
Yin, P	2021 [24]	PS	LSCS	PETLIF	TF	56	60.5	10/46	204.2	general/epi	105.6	-	15.3	direct	cage → PPS
				PLIF	-	58	60.6	10/48	99.8	general/epi	241.6	-	15.8	direct	
Zhang, H	2021 [25]	RS	DLS	Endo-TLIF	TF	32	53.1	12/20	202.6	general	73.0	1.6	>12	direct	cage → PPS
				MIS-TLIF	-	30	55.7	14/16	192.1	general	129.0	2.3	>12	direct	
Jin, M	2020 [26]	RS		PELIF	TF	16	61.2	9/7	155.2	local	35.0	4.9	22.6	direct	cage → PPS
				OLIF	-	32	63.9	18/14	160.3	general	80.0	4.3	24.2	indirect	
Ao, S	2020 [27]	PS	DLS, LDH, LSCS	PETLIF	TF	35	52.8	16/19	143.0	general	492.7	3.1	>14	?	cage → PPS
				MIS-TLIF	-	40	53.6	22/18	103.6	general	698.1	5.2	>14	direct	
Yang, J	2019 [28]	RS	LSCS	PE-TLIF	TF	7	55.3	1/6	285.7	general/epi/local	117.1	4.0	>12	direct	cage → PPS
Cheng, X	2023 [29]	RS	DLS	PE-TLIF	TF	27	70.7	22/5	157.59	local	47.41	7.85	>12	direct	PPS → cage

RS; retrospective study, PS prospective study, LSCS; lumbar spinal canal stenosis, DLS; degenerative lumbar spondylolisthesis, LDH; lumbar disc herniation, LDD; lumbar disc disease, IL; Interlaminar, TF; Transfroaminal, PE; percutaneous endoscopic, LIF; lumbar interbody fusion, PLIF; posterior LIF, MISTLIF; minimally invasive surgery transforaminal LIF, OLIF; oblique LIF, ETLIF; endoscopic TLIF, epi; epidural, PPS; percutaneous pedicle screw.

**Table 2 jcm-12-05391-t002:** Patients and surgical characteristics.

Characteristic	Value
Case number	24
Age (years)	
Mean	70.5 ± 2.19
Range	51–86
Sex	
Male	13
Female	11
Diagnosis	
Degenerative spondylolisthesis	7
Lumbar spinal canal stenosis	17
Surgical levels	
L2–3	1
L3–4	9
L4–5	14
L5–S1	0
Operation time (mins)	130.8 ± 9.2
Intraoperative blood loss (mL)	24.0 ± 4.9
Postoperative hospitalization time (days)	9.4 ± 0.6
Follow up periods (month)	21.5 ± 0.5

**Table 3 jcm-12-05391-t003:** Patients’ data.

Case No.	Diagnosis	Surgical Procedure	Endplate Injury	Cage Subsidence	Exiting Nerve Injury	Bone Fusion
1	Degenerative spondylolisthesis	PETLIF	−	+	−	+
2	Lumbar spinal canal stenosis	PETLIF	−	+	−	+
3	Lumbar spinal canal stenosis	PETLIF	−	−	−	+
4	Lumbar spinal canal stenosis	PETLIF	−	+	−	+
5	Lumbar spinal canal stenosis	PETLIF	−	+	−	−
6	Lumbar spinal canal stenosis	PETLIF	−	−	−	+
7	Lumbar spinal canal stenosis	PETLIF	−	−	−	−
8	Lumbar spinal canal stenosis	PETLIF	−	+	−	+
9	Degenerative spondylolisthesis	PETLIF	−	+	−	+
10	Degenerative spondylolisthesis	PETLIF	−	−	+	+
11	Degenerative spondylolisthesis	PETLIF	−	−	−	−
12	Degenerative spondylolisthesis	PETLIF	+	−	−	+
13	Degenerative spondylolisthesis	PETLIF	−	−	−	+
14	Degenerative spondylolisthesis	PETLIF	−	−	−	+
15	Degenerative spondylolisthesis	PETLIF	−	+	−	+
16	Degenerative spondylolisthesis	PETLIF	−	−	−	+
17	Degenerative spondylolisthesis	PETLIF	−	−	−	+
18	Degenerative spondylolisthesis	PETLIF	−	−	−	+
19	Degenerative spondylolisthesis	PETLIF	−	−	−	+
20	Degenerative spondylolisthesis	PETLIF	−	−	+	+
21	Degenerative spondylolisthesis	PETLIF	−	−	−	+
22	Degenerative spondylolisthesis	PETLIF	−	−	−	+
23	Degenerative spondylolisthesis	PETLIF	−	−	−	+
24	Degenerative spondylolisthesis	PETLIF	−	−	−	+

**Table 4 jcm-12-05391-t004:** Clinical outcomes (JOA, VAS).

Scores	Preop	1M	3M	6M	Final
JOA score	17.4 ± 0.8	25.7 ± 0.4 *	27.0 ± 0.4 *	26.9 ± 0.3 *	26.9 ± 0.3 *
VAS of low back pain	51.0 ± 5.1	15.6 ± 4.2 *	7.3 ± 2.6 *	6.9 ± 2.1 *	6.5 ± 2.0 *
VAS of leg pain	74.4 ± 3.7	10.8 ± 2.9 *	11.7 ± 3.6 *	9.5 ± 2.6 *	10.8 ± 3.5 *

* Significantly different from the preoperative value (*p* < 0.05).

**Table 5 jcm-12-05391-t005:** Radiological data.

Parameters	Preop	1M	3M	6M	Final
Lumbar lordosis (LL)	33.3	32.6	33.4	33.0	33.3
Pelvic incidence (PI)	48.9	48.9	48.9	48.9	48.9
PI-LL	15.6	16.3	15.5	15.9	15.6
Local lordosis	9.3	10.0	9.9	9.2	9.0
% Slip	17.3	n.a.	n.a.	n.a.	4.5 *
Disc height	5.1	n.a.	n.a.	n.a.	7.3 *
Spinal canal area	43.1	n.a.	n.a.	n.a.	108.8 *

* Significantly different from the preoperative value (*p* < 0.05).

## Data Availability

The data used to support the funding of this study are available from the corresponding author upon request.

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
