# Peer review of "Percutaneous Endoscopic Transforaminal Lumbar Interbody Fusion (PETLIF): Current Techniques, Clinical Outcomes, and Narrative Review"

_jcm, 2023, doi:10.3390/jcm12165391_

Round 1
Reviewer 1 Report
Figure 2: each incision should be marked.
Figure 3: the cranial side and anatomic landmark should be marked.
Figure 5: the cage was over-punched into the contralateral side.
table 4: the bone union did not belong to the complication.
study design: this study lack the control group.
The English writing is good.
Author Response
Dear Reviewer1,
Thank you for inviting us to submit a revised draft of our manuscript. We also appreciate the time and effort you have dedicated to providing insightful feedback on ways to strengthen our paper. Thus, it is with great pleasure that we resubmit our article for further consideration. We have incorporated changes that reflect the detailed suggestions you have provided. We also hope that our edits and responses below satisfactorily address all the issues and concerns you and the reviewers have noted.
Figure 2: each incision should be marked.
Thank you for raising this point. We have added marks on incisions as PPS entry points (p), posterior-lateral entry for the cage (※), and incision for bone harvesting (△).
Figure 3: the cranial side and anatomic landmark should be marked.
The cranial side is indicated by text to the figure's left, and anatomical landmarks are marked in the figure and added to the description in red. Figure 3 has been changed to Figure 4 due to the addition of Figure 2, as pointed out by the other reviewer.
Figure 5: the cage was over-punched into the contralateral side.
The cage was placed on a mechanically robust rim to avoid subsidence. We have added the sentence in the "2.4.8 cage insertion" paragraph.
Table 4: the bone union did not belong to the complication.
We have reworked Table 4, adding the points raised by the other reviewers, and changed to Table 3, Patient data.
Study design: this study lacks a control group.
This is an important point, but unfortunately, we do not have a control group in this study.

Reviewer 2 Report
Authors present a retrospective case series of 24 patients who underwent percutaneous endoscopic transforaminal lumbar interbody fusion (PETLIF) and narrative review. In the title there is a mistake - if you talk about narrative review and then present a PRISMA flow chart, this cannot be combined - either you performed a systematic review using PRISMA or narrative review, where you comment selected papers on a certain subject. One important recent study has been left out of the literature review:
Cheng X, Yan H, Chen B, Tang J. Percutaneous pedicle screw fixation with percutaneous endoscopic transforaminal lumbar interbody fusion in the treatment of degenerative lumbar spondylolisthesis with instability. World Neurosurg. 2023 Jun 9:S1878-8750(23)00777-5. doi: 10.1016/j.wneu.2023.06.012. Epub ahead of print. PMID: 37302710.
Surgical technique is nicely described. Please include a Table with all patients and their characteristics - indications, surgical treatment. Include at least three illustrative cases with relevant imaging. Please include in the illustrative cases a discussion on alternative therapy to PETLIF - MIS TLIF, open technique; XLIF, prone XLIF. Discuss contraindications to PETLIF and endoscopy general, and when do you need to convert the surgery from endoscopic to open (CSF leak etc).
2.4..4 - please provide a photo of the entry point;
If possible provide an operative video and obligatory photo of operative setting - where is the surgeon position to patient, camera position, endoscope, C-arm etc.
In the Results - first provide data on your patients. I suggest to include literature review separately.
Report complication rate and clinical outcome of the patients. Illustrative cases which you provided - were they treated conservatively and for how long? What was the hospital stay? Ambulatory surgery?
Acceptable.
Author Response
Dear Reviewer2,
Thank you for inviting us to submit a revised draft of our manuscript. We also appreciate the time and effort you have dedicated to providing insightful feedback on ways to strengthen our paper. Thus, it is with great pleasure that we resubmit our article for further consideration. We have incorporated changes that reflect the detailed suggestions you have provided. We also hope that our edits and responses below satisfactorily address all the issues and concerns you and the reviewers have noted.
Comments and Suggestions for Authors
The authors present a retrospective case series of 24 patients who underwent percutaneous endoscopic transforaminal lumbar interbody fusion (PETLIF) and narrative review. In the title, there is a mistake - if you talk about narrative review and then present a PRISMA flow chart, this cannot be combined - either you performed a systematic review using PRISMA or narrative review, where you comment selected papers on a particular subject. One important recent study has been left out of the literature review:
The word PRISMA has been removed for narrative review. However, we collected and evaluated articles selected using specific criteria to compare how PETLIF differs from previous endoscopic lumbar interbody fusion. We have added the suggested papers to Figure 1. and Table 1.
The surgical technique is nicely described. Please include a Table with all patients and their characteristics - indications, surgical treatment. Include at least three illustrative cases with relevant imaging. Please include in the illustrative cases a discussion on alternative therapy to PETLIF - MIS TLIF, open technique; XLIF, prone XLIF. Discuss contraindications to PETLIF and endoscopy general, and when do you need to convert the surgery from endoscopic to open (CSF leak etc).
We have included Figure 9 as the other case 2 and discussed the contraindications of PETLIF and alternative techniques in the 4. Discussion paragraph.
2.4..4 - please provide a photo of the entry point;
We have shown the posterior-lateral entry for the cage in Figure 3 (a) as ※.
If possible provide an operative video and obligatory photo of operative setting - where is the surgeon position to patient, camera position, endoscope, C-arm etc.
The obligatory photo of the operative setting was added as Figure 2, and a description was inserted in 2.4.1. Preparation and planning paragraph. And we have an operative video, but we cannot provide the video due to ethical issues.
In the Results - first provide data on your patients. I suggest to include literature review separately.
The literature review in the results section has been partially moved to the literature review section in the materials and methods shown in red. And a new paragraph, “3. Clinical data from the previous literature,” was created to analyze the previous literature. And the result of our data was moved to 4—the resulting paragraph.
Report complication rate and clinical outcome of the patients. Illustrative cases which you provided - were they treated conservatively and for how long? What was the hospital stay? Ambulatory surgery?
Clinical results are summarized in TABLE 4. The complication rates are listed in red in paragraph 4.2. All the cases of cage subsidence were not clinically problematic as they eventually acquired bone fusion. Two cases of exiting nerve injury were described in detail, although they eventually improved. One case stayed 21 days after the surgery, whereas the other left the hospital on day 8.

Reviewer 3 Report
The topic is of high interest.
The authors should present their fusion rate, and what was done if there were cases with pseudarthrosis.
The authors should present in detail their 2 nerve injuries. What were the symptoms? did they improve over time?
How many surgeons were involved and how experienced were they in the endoscopic techniques?
What kind of iliac crest harvester was used?
English can be improved, particularly in the description of the surgical technique.
Author Response
Dear Reviewer 3,
Thank you for inviting us to submit a revised draft of our manuscript. We also appreciate the time and effort you have dedicated to providing insightful feedback on ways to strengthen our paper. Thus, it is with great pleasure that we resubmit our article for further consideration. We have incorporated changes that reflect the detailed suggestions you have provided. We also hope that our edits and responses below satisfactorily address all the issues and concerns you and the reviewers have noted.
Comments and Suggestions for Authors
The topic is of high interest.
The authors should present their fusion rate, and what was done if there were cases with pseudarthrosis.
Thank you for raising this critical point. Regarding bone fusion, we used the term “bone union” and presented in Table 4. Among 24 cases, 21 cases have achieved bone fusion. The rate was 87.5%. We have changed the word to “bone fusion” in the table and added a sentence explaining the bone fusion rate in the “4.1 Patients’ characteristics and clinical outcomes” paragraph in red.
Of the three cases of non-fusion, one had bone fusion at follow-up after completing this manuscript. Two of the three patients are still under outpatient observation while continuing to receive teriparatide because their pain is under control. In red, we have added this sentence in the “3.2 Patients’ characteristics and clinical outcomes” paragraph.
The authors should present in detail their 2 nerve injuries. What were the symptoms? did they improve over time?
Exiting nerve injury occurred in cases no.10 and 20. The recovery time was four months in case 10 and 5 months in case 20. In case 10, the patient experienced lower limb fatigue on the cage entry side after walking long distances. In case 20, muscle strength was affected, such as the disability to ascend and descend stairs, but these symptoms improved along with the pain. We have added the sentence in the “4.2 Complication” paragraph.

Round 2
Reviewer 2 Report
Authors have sufficiently responded to remarks.
Acceptable.